# Honey contamination from plant protection products approved for cocoa (Theobroma cacao) cultivation: A systematic review of existing research and methods

**Richard G. Boakye**[1,2]*, **Dara A. Stanley**[1,2], **Blanaid White**[3,4]

**1** School of Agriculture and Food Science, University College Dublin, Dublin, Ireland, **2** Earth Institute, University College Dublin, Belfield, Dublin, Ireland, **3** School of Chemical Sciences, Dublin City University, Dublin, Ireland, **4** National Centre for Sensor Research, DCU Water Institute, Dublin City University, Dublin, Ireland

* rgboakye@yahoo.co.uk

## Abstract

The main component of chocolate, cocoa (Theobroma cacao), is a significant commercial agricultural plant that directly sustains the livelihoods of an estimated forty to fifty million people. The economies of many cocoa producing nations, particularly those in the developing world, are supported by cocoa export revenue. To ensure satisfactory yields, however, the plant is usually intensely treated with pesticides because it is vulnerable to disease and pest attacks. Even though pesticides help protect the cocoa plant, unintended environmental contamination is also likely. Honey, produced from nectar obtained by honeybees from flowers while foraging, can serve as a good indicator for the level of pesticide residues and environmental pesticide build-up in landscapes. Here, we use a systematic literature review to quantify the extent of research on residues of pesticides used in cocoa cultivation in honey. In 81% of the 104 studies examined for this analysis, 169 distinct compounds were detected. Imidacloprid was the most frequently detected pesticide, making neonicotinoids the most frequently found class of pesticides overall. However, in cocoa producing countries, organophosphates, organochlorines, and pyrethroids were the most frequently detected pesticides. Interestingly, only 19% of studies were carried out in cocoa producing countries. We recommend prioritizing more research in the countries that produce cocoa to help to understand the potential impact of pesticide residues linked with cocoa cultivation in honey and the environment more generally to inform better pesticide usage, human health, and environmental policies.

## 1. Introduction

Cocoa (Theobroma cacao) is an essential economic crop with widespread global demand and uses owing to its rich protein, carbohydrate, fat and vitamin content [1]. Between forty and fifty million people are thought to rely on cocoa farming for a living [2, 3]. The annual global

**Data Availability Statement:** All relevant data are within the paper and its Supporting Information files

**Funding:** RGB. Project ID: GOIPG/2019/517. The study was funded by the Irish Research Council. The funders had no role in study design, data collection and analysis, decision to publish, or preparation of the manuscript. URLs: https://research.ie/.

**Competing interests:** The authors have declared that no competing interests exist.

production of 4.2 million metric tons of cocoa beans is estimated to have an economic value of over US$11.8 billion [4]. The cocoa confectionary market generates about US$80 billion worldwide [5], with West Africa as its primary production engine. However, the cocoa plant is susceptible [6, 7] to attacks from the cocoa swollen shoot virus, beetles and capsids (miridae) and phytophthora pod rot (commonly called black pod) [8, 9]. This results in the loss of 20% to 30% of the cocoa produced worldwide [10], valued at US$3.16 billion [11].

To mitigate the impact of disease and pest pressure, [12] pesticides are widely used in cocoa cultivation [13, 14]. For example, an estimated 125,000–130,000 metric tons of insecticides are applied for cocoa cultivation in Nigeria alone [15]. Concerns have been expressed regarding the potential environmental damage caused by pesticides, as well as implications for residues in food [16], considering that only 0.01% of applied pesticides are determined to reach their targets while the rest filters into the broader environmental ecosystem [17–20].

Honey, created when stingless bees and honeybees collect nectar and/or other resources from flowers and plants [21, 22], is used by bees as a nutritious food source [21, 23]. While bees are foraging for nectar to make honey, they can also collect potential environmental contaminants, including pesticides [24]. Because honey may reflect the chemical conditions of the environment, it can be used as a proxy to assess general ecosystem health [25, 26]. For example, honey and other bee products have been used to assess environmental contaminants, including heavy metals [27, 28], polycyclic aromatic hydrocarbons [29] and pesticides [30, 31].

Pesticide residues in honey can be related to their potential impact on human health using maximum residue limits (MRLs). An MRL (expressed as the milligram of residue per kilogram of feed commodity) is the highest permissible pesticide residue recommended by the Codex Alimentarius Commission as legally accepted in food commodities and animal feeds [32]. When determining MRLs, the European Union, one of the world's largest agricultural product markets, considers Codex Alimentarius requirements and good agricultural practices [33]. The European Food Safety Authority (EFSA) calculates MRLs, assuring compliance with globally recognized assessment techniques [34]. As a natural food produced by *Apis mellifera*, honey is deemed a food substance of animal origin under Directive 2001/110/EC and therefore needs to meet specified MRL requirements. Unfortunately, different national or regional bodies may set different upper pesticide residue concentration limits, which may lead to confusion in international markets [35]. Therefore, MRL harmonisation and standardization is essential.

This review utilizes the relationship between pesticide contamination of honey and pesticide use in the broader environment to evaluate the current knowledge of residues of pesticides used for cocoa cultivation in honey. Specifically, the study aimed to:

- Analyse the time-frame and geographic location of previous studies of honey contamination by pesticides permitted for cocoa cultivation.

- Investigate the extent to which various pesticide classes and varieties have been reported in cocoa producing countries.

- Evaluate the potential impact of the pesticides reported for human health utilising the European Union maximum residue limits (MRL).

## 2. Materials and methods

### 2.1. Pesticides utilised for cocoa cultivation

Seven peer-reviewed publications and one international report were identified that contain data on the approved pesticides for cocoa cultivation in four major cocoa producing nations: Ghana, Nigeria, Cameroon and Ivory Coast, which account for 70% of the world's cocoa

**Table 1. Summary of approved active ingredients for cocoa production.** These pesticides are recommended for cocoa cultivation in major cocoa producing countries in West Africa which account for 70% of the World's cocoa.

| Insecticides (active) ingredients | Fungicides | Herbicides | Sources |
|---|---|---|---|
| Acetaprimid | Benalaxyl | Glyphosate | [9, 37–43] |
| Bifenthrin | Benomyl | Paraquat | |
| Capsaicin | Copper (II) hydroxide | | |
| Chlorantraniliprole Chlorpyriphos | Copper (I) hydroxide | | |
| Lambda-Cyhalothrin | Copper (I) oxide | | |
| Alpha-Cypermethrin Cypermethrin | Dicopper chloride trihydroxide | | |
| Deltamethrin | Dimethomorph | | |
| Dimethoate | Fluazinam | | |
| Etofenprox | Maned | | |
| Fipronil | Mancozeb | | |
| Imidacloprid | Mefenoxam | | |
| Indoxacarb | Metalaxyl | | |
| Pirimiphosmethyl | Metalaxyl-M | | |
| Promecarb | | | |
| Pyrethrum | | | |
| Sulfoxaflor | | | |
| Teflubenzuron | | | |
| Thiamethoxam | | | |

production [36]. Using these publications, a pesticide list was compiled of the most important pesticides for cocoa, which included twenty-three insecticides, seventeen fungicides, and two herbicides approved for cocoa growing (Table 1).

## 2.2. Formulation of search strings

A collection of set-specific search strings was created to systematically search the literature for studies evaluating Table 1 pesticides in honey. Each string included a term for honey, plus a list of some of the pesticides of interest and was divided into three: one for insecticides, one for fungicides and one for herbicides (S1 Table). To make the search string for the insecticides shorter for the search engine, it was further split into two parts.

## 2.3. Literature search

The search strings created were utilized to conduct searches through the Web of Science Core Collection, PubMed, and Scopus. An initial search was conducted on 12[th] October 2020, which resulted in 1,360 peer-reviewed studies from Web of Science and PubMed, while a further search was conducted in Scopus on 26[th] October 2020, which resulted in the retrieval of 524 studies. Books or book portions, theses, and grey literature were excluded [44], as well as any study in languages other than English. After removing duplicates, 1,282 studies were screened based on titles and abstracts for the presence of pesticide active ingredient residues in honey, which produced 91 studies in total. One paper was inaccessible, so 90 studies progressed to quality review. The flow chart and PRISMA table in Fig 1 and S2 Table, based on [45], show the procedure taken to arrive at the included studies at the start of this review. A supplementary data search was also carried out on 7th November 2022 for literature released between November 2020 and November 2022 to ensure more recent literature was also captured. The updated flow chart is shown in S1 Fig. After the initial list of 2,610 studies was processed as described above, an additional 23 studies satisfied the eligibility criteria for inclusion. Before text screening and quality assessment, the publication by [46] was omitted because it was inaccessible. This resulted in an additional 22 papers of the updated search progressing to the quality review stage.

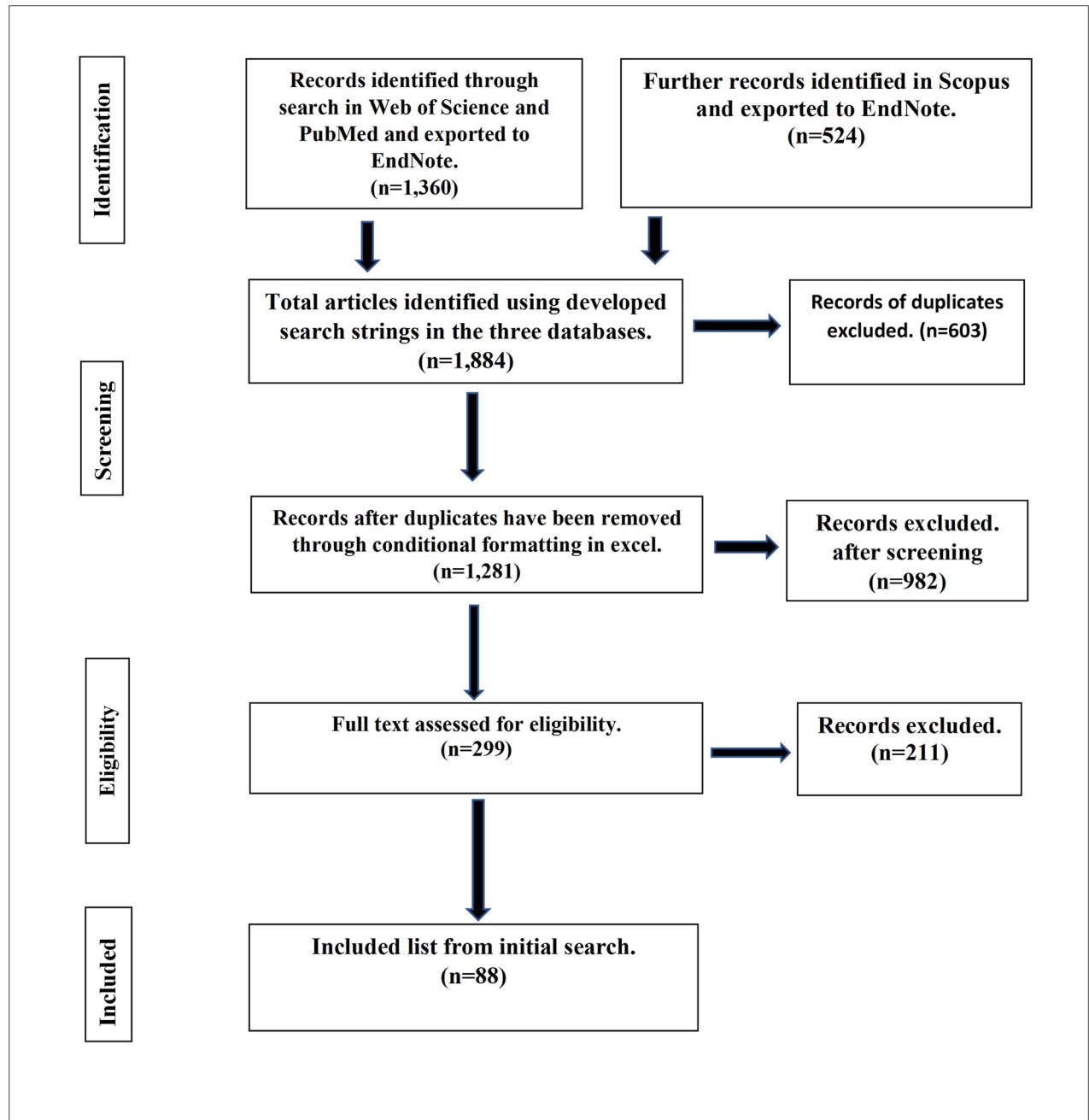

**Fig 1. Procedure followed to select studies for inclusion in this systematic literature review.** Numbers presented are from the first search of the literature in November 2020 only, while results from the supplementary search in November 2022 are given in S1 Fig. The "eligibility" box covers both the eligibility screen and quality assessment. Based on PRISMA flow chart [45].

## 2.4. Quality assessment

Overall, 112 studies from the previous and updated searches were proceeded to quality assessment, and applicability before data extraction (S2 Table). We applied a checklist of eleven customized questions (S1 Appendix) based on the proposed checklists for evaluating quantitative studies by [47] to each study. Two reviewers assessed each paper using a scoring methodology [47]. According to this grading scale, particular studies were given a score based on how much

they met the criteria (yes" = 2, "partial" = 1, "no" = 0, NA = not applicable). The reviewers' consensus on the total scores ranged from 45% to 100%. Eight studies were excluded based on quality assessment results. Three of these eight papers could not be attributed explicitly to one nation for examination. The other five did not measure the levels of pesticide residues in samples, instead using blank honey as a sample matrix to show the analytical method's reliability. 104 studies were determined to have passed the review's quality assessment (S2 Table). Table 2 summarizes the categories of data extracted from each paper, including information on the publication year, study location, types of pesticides examined and found, and data extraction and analytical methods. The final dataset is presented in S3 Table. A brief descriptive summary of the main findings of each study included in our systematic review was determined from information provided within each paper and collated for qualitative purposes (S4 Table).

Pesticides were recorded in three categories: insecticides, fungicides, and herbicides. Pesticides targeted in each study were recorded, and those detected were subsequently identified and their concentrations recorded. Their concentrations were then compared with the maximum residue limits (MRL) set by the European Union to determine those which exceeded the MRL. All units of concentrations of detected pesticides were converted to mg/kg before comparison. To evaluate the sensitivity of the analytical technique used in each investigation, LOQs applied were compared to compound MRLs.

## 3. Results

### 3.1. Geographical spread and period of study

The first study was published in 1997, but it was in 2015 that study publications began to increase, with 73% of studies taking place between 2015 and 2022 (Fig 2). The geographical locations and the period within which the studies were conducted are documented in S3 Table.

Most studies took place in Asia (30%) and Europe (43%). One third of the 47 studies, that were carried out in Europe, took place in Spain. Similarly, of the 31 studies conducted in Asia, 48% were conducted in China. Overall, 81% of the included studies were conducted in 27 countries where cocoa is not grown (Fig 3).

There are 57 cocoa producing countries globally (Fig 4), but studies were only conducted in eight. There is an uneven distribution of studies across these eight countries. Of the 20 studies conducted in cocoa producing countries, 8 studies were carried out in Brazil, the sixth-highest cocoa producer in the world, accounting for 5% of the world's cocoa bean production (S5 Table). Only one publication was carried out in each of the Ivory Coast and Ghana, rated first and second with 39% and 17% of the yearly global cocoa production. In contrast, four studies were conducted in India, which accounts for less than 1% of global annual production. The other nations producing cocoa where studies took place included Mexico (2 studies), Pakistan (2 studies), Thailand (1 research), and Uganda (1 study).

### 3.2. The classes and types of pesticides evaluated

Among the classes of pesticides investigated, insecticides received the most research attention, having been examined in 91% of studies. Only four studies examined insecticides, fungicides, and herbicides simultaneously. Pesticide traces were found in 80% of publications included in this review, with a total of 169 different compounds (comprising some of those recommended as well as those not approved for cocoa cultivation) detected in 86 studies, which took place in 30 of the 35 countries where studies were conducted (Fig 5).

Neonicotinoids were both the pesticide classes most investigated and with the greatest detections overall, with imidacloprid (detected in 20 studies), thiamethoxam (detected in 14

**Table 2. Type of data extracted from included articles.** Based on this, a customised data extraction form was developed and used for data capture and processing.

| Category | Sub-category |
|---|---|
| General Reference Information | Author lists and correspondent's contact details |
| | Title of paper |
| | Journal name/source |
| | Volume |
| | Issue |
| | Pages of journal |
| | Year of publication |
| | DOI |
| Method | Aim of study |
| | Study design |
| | Start date |
| | End date |
| | No of samples |
| | Weight of sample |
| | Volume analysed |
| | Extraction technique |
| | Analytical technique |
| | Statistical analysis |
| Geographical locations of the study | Country name |
| | Continent |
| | Country status |
| Pesticide studied | What pesticides were studied |
| | Were insecticides studied? |
| | Were fungicides studied? |
| | Were herbicides detected? |
| | Concentrations of pesticide residues detected |
| | Banned pesticides detected |
| Analytical parameters | Level of detection |
| | Level of quantification |
| | MRL of detected pesticides |
| Type of honey analysed | Unifloral |
| | Multi-floral |
| Source of honey analysed | Commercial honey only |
| | Commercial honey and honey directly harvested |
| | Honey harvested directly from production base |
| Sample treatment | Heated |
| | Pasteurized |
| Season when sample was taken | |
| Number of times/Seasons samples taken for analysis | Full season (multiple harvest) |
| | Part season (single harvest) |
| Matrices analysed | Honey only |
| | Honey and other matrices |
| | For multiple harvest: Did the concentration fluctuate between studies? |
| Honey treatment | Pasteurized |
| | Blended |

(*Continued*)

**Table 2.** (Continued)

| Category | Sub-category |
|---|---|
| Honey characteristics | Electrical conductivity |
| | % Water |
| | Colour |
| Highlights | Key results from the study |
| | Summary of abstract |

studies), acetamiprid (detected in 13 studies) and clothianidin (detected in 9 studies) being the more commonly detected neonicotinoids. Of the eight cocoa producing countries, pesticides were only detected in 6 of these countries, with no detections in Thailand or Uganda, and interestingly, the three most detected pesticide classes in the six cocoa-growing countries were organophosphates, organochlorines and pyrethroids in that order (Fig 6 and S6 Table). Eleven approved insecticides for cocoa cultivation, namely capsaicin, chlorantraniliprole, thiamethoxam, acetaprimid, etofenprox, indoxacarb, pirimiphosmethyl, promecarb, pyrethrum, sulfoxaflor, and teflubenzuron and one herbicide (i.e. paraquat) were not detected in any of the studies conducted in the cocoa growing countries. Additionally, our findings showed that only two of the 13 recommended fungicides for cocoa production, namely metalaxyl-M and its isomer metalaxyl (S7 Table) were detected in studies conducted in cocoa growing countries. Forty-nine pesticides were detected in studies undertaken in Brazil, Mexico, and India that are not suggested for use in the production of cocoa [50]. However, it should be remembered that whilst cocoa production occurs in these countries, it was not possible to uniformly ascertain whether the honey samples analysed in the studies were collected from cocoa producing regions within these countries.

Multiple studies were conducted in just over half (54%) of the thirty-five countries where studies took place. Fifteen studies were carried out in Spain and China. There was a far lower incidence of multiple studies in cocoa producing countries, with only Brazil, India, and Mexico having more than one study conducted. Among the cocoa growing countries, Brazil was the only country where a pesticide (i.e., chlorpyrifos) was detected in different studies. Overall, only seven different pesticides were detected on multiple occasion in all multiple studies conducted across all countries (S7 Table).

**3.2.1. Banned pesticides detected.** Various jurisdictions have made the use of certain pesticides illegal. Using the Stockholm Convention as the foundation for evaluating "banned" pesticides, 96% of the studies considered in this review did not detect any banned chemicals. Banned pesticides were detected only in three cocoa producing countries (Ghana, India, and Mexico) and were predominantly organochlorines. One study from a non-cocoa producing country, Spain, also confirmed the detection of banned pesticides. In the study conducted in Ghana, Dichlorodiphenyltrichloroethane (DDT), an organochlorine insecticide which is the first of the modern synthetic insecticide manufactured primarily to fight malaria, typhus and for agricultural uses [51], was confirmed at 0.01 mg/kg concentration. In Mexico, [52] confirmed the presence of 10 organochlorines, including heptachlor (0.13173 mg/kg); hexachlorocyclohexane (HCF, 0.654 mg/kg), endrin aldehyde (0.03564 mg/kg), and dichlorodiphenyldichloroethylene (DDE, 0.154358 mg/kg) in honey from the Chiapas vicinity where official approval for their usage was withdrawn in the 2000s, prior to the study being carried out. In India, hexachlorocyclohexane (HCH), which is used as an insecticide on fruits, vegetables, and forest crops, and its isomers, endosulfan and aldrin, were detected at concentrations of 0.0028 mg/kg, 0.00253 mg/kg, and 0.00201 mg/kg, respectively. However, it must

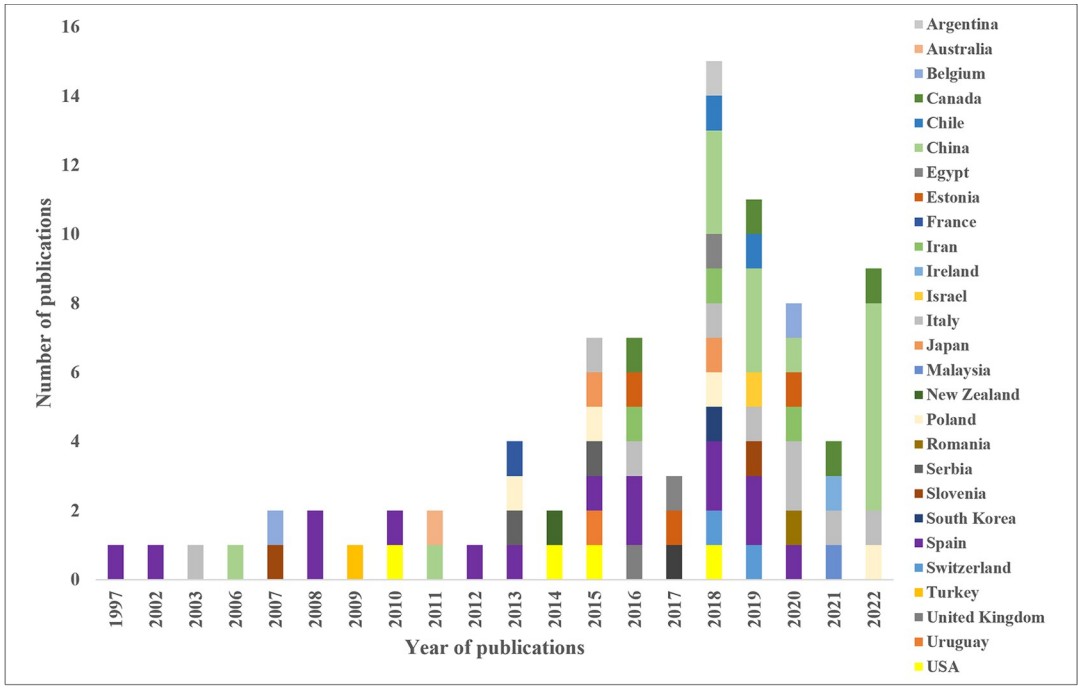

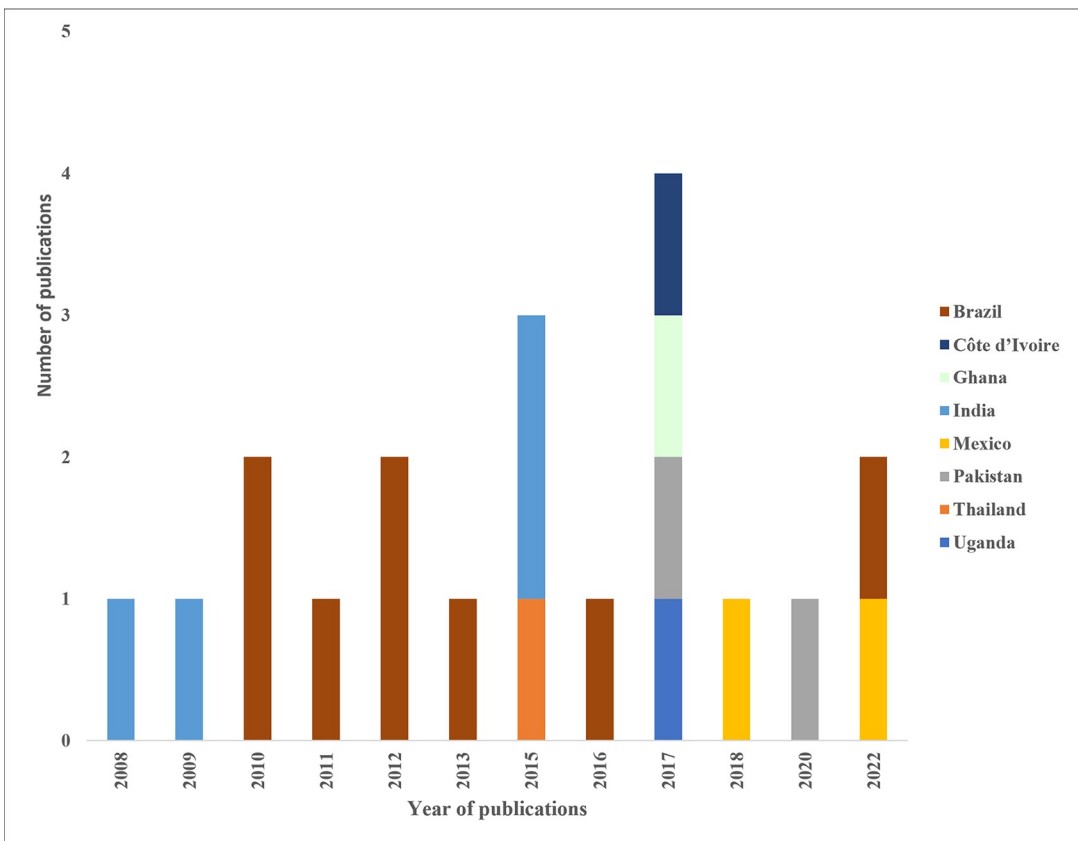

**Fig 2. The year and country where existing studies were conducted.** The studies were published over 25 years across 35 countries. Above) Non-cocoa growing countries where studies were conducted. Below) Cocoa producing countries where studies took place. Overall, 18 studies were conducted in eight cocoa growing countries.

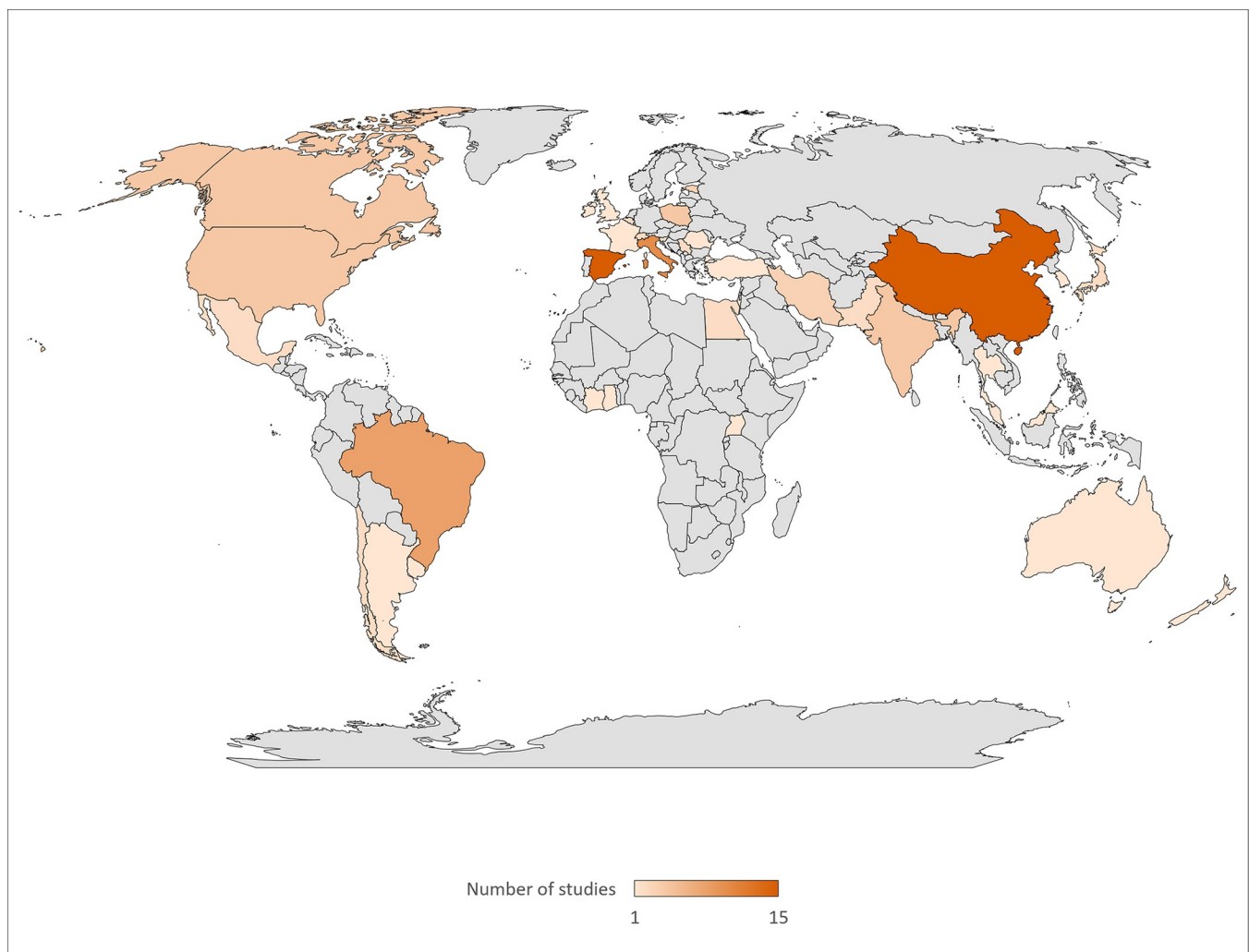

**Fig 3. Geographical spread of the studies that were undertaken (grey areas = no studies).** Spain (15 studies), China (15 studies) and Italy (10 studies) were the top three countries with most studies. One study each was conducted in Ivory Coast and Ghana, which are the first and second ranked cocoa producing counties in the world, respectively. [Credit: [48]].

be noted that DDT is still permitted in India for fumigation against mosquitoes as a malaria control tool, unlike the other three countries where it was detected [53]. One of the sixteen studies in Spain found DDE (0.09–0.6598 mg/kg), a metabolite of DDT, to be the only banned pesticide detected.

## 3.3. Exceedance of maximum residue limits

In all, 12% of studies reported pesticide residue quantities in honey that exceeded the MRL established by the European Union (Table 3). These studies were conducted in ten different countries. EU MRLs are occasionally revised in light of additional scientific data becoming available to the European Food Safety Authority. During the period of this systematic review, these revisions resulted in an increase in MRL for certain specific pesticides and the 2022 MRLs are the primary focus here while the 2005 ones are also reported in Table 3. Among the cocoa producing counties, MRLs exceedances occurred in Brazil, Ivory Coast, and India. However, in India, malathion levels only exceeded MRL set by India; the concentration observed

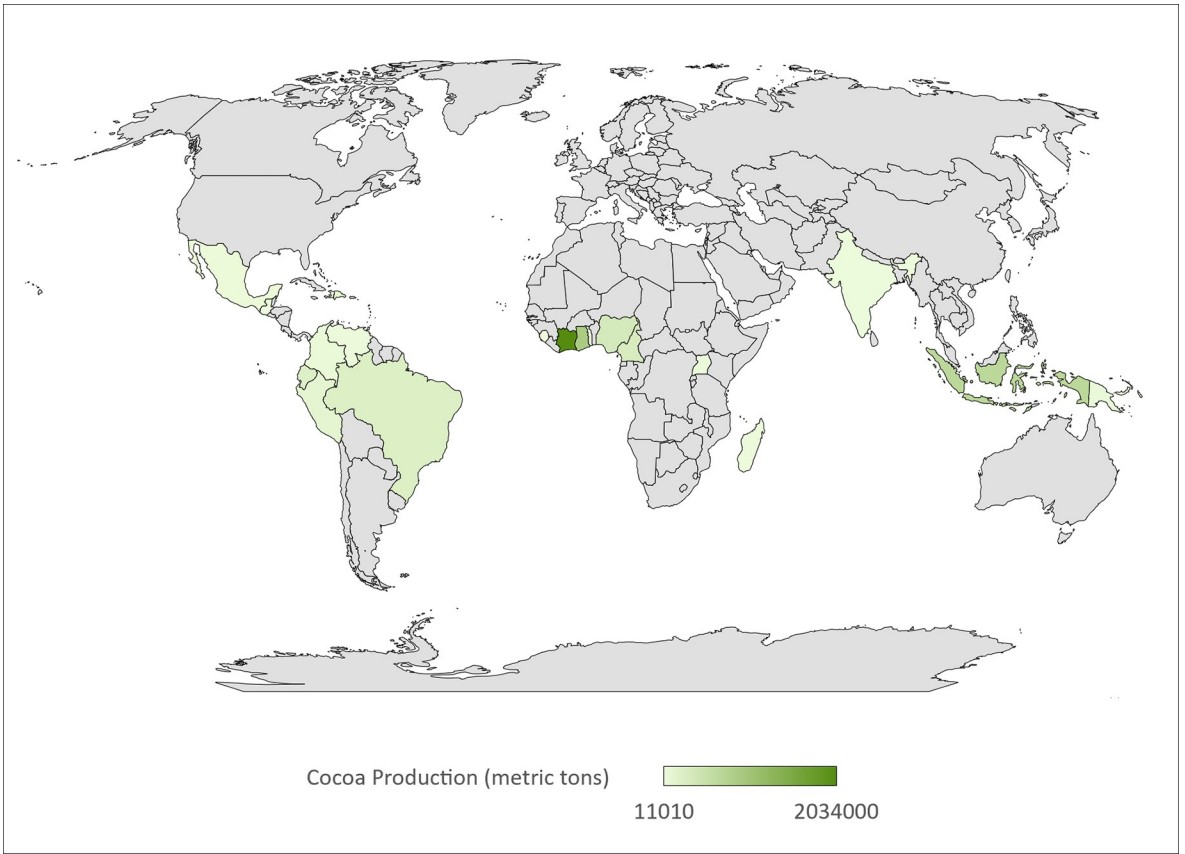

**Fig 4. The 57 cocoa producing countries in the world based on the metric tons of cocoa produced annually.** Ivory Coast is ranked first with 2,034,000 metric tons of annual production. 19% of the included studies evaluated in this review took place in eight cocoa producing countries. Grey areas = non-cocoa producing areas. Map developed using [48] and data sourced from [49].

was lower than the EU's previous and current MRL. No exceedances of MRLs were detected in North America (4 studies) or Australia (1 study).

## 3.4. The types of honey investigated in existing studies

Honey can be obtained in two forms; raw straight from the hive [64] or processed commercially, including heating and cooling to lower moisture content [65]. Except for one study done in Spain, where the source of the honey analyzed was not indicated, both commercial honeys (35 studies) and raw honeys (61 studies) were analysed in the included studies, with both being analyzed simultaneously in 7%. Although it has been established that heating tends to decrease honey quality with the potential to degrade pesticide residues [66], it was not possible to assess how this may have affected the levels of pesticides because information on honey's prior heating or pasteurization was not frequently recorded for in-depth analysis. None of the studies included in our study indicated whether they used blended honey.

**3.4.1. Honey sampling rate within studies.** Most of the research analyzed pesticide residues in honeys sampled only once. Only 11 studies repeatedly collected and examined honey samples for pesticide contamination, all using raw honey, except in one study conducted in Uganda, where commercial honey was used. In eight of these studies, honey samples were gathered and examined over two years or several months within a single year. A unique study analysed honey samples continuously for nine years in Estonia. No trends emerged in studies

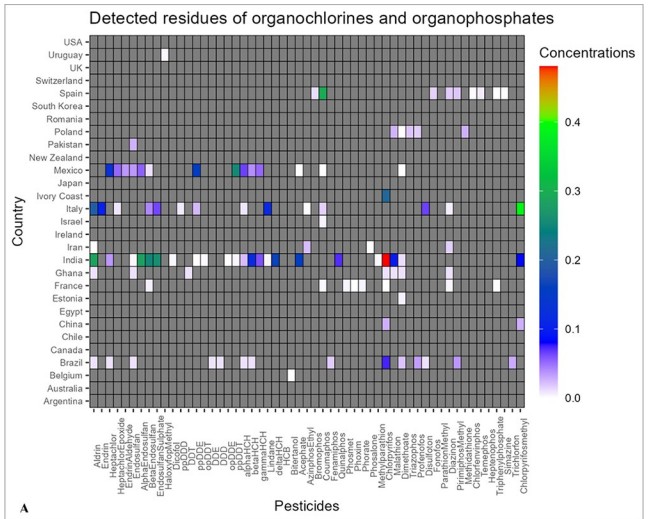

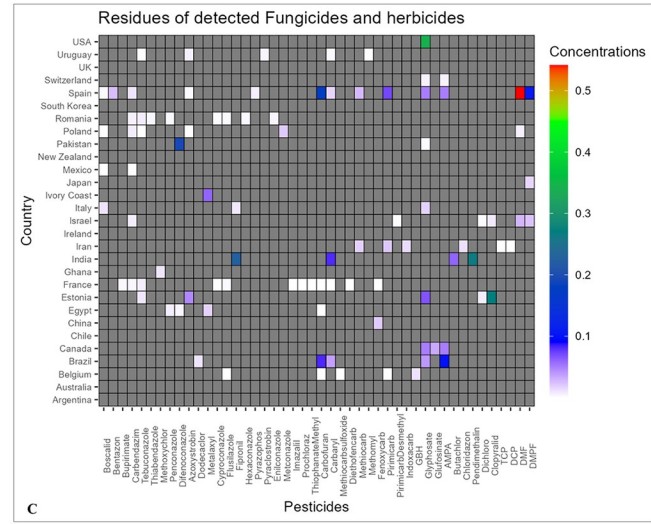

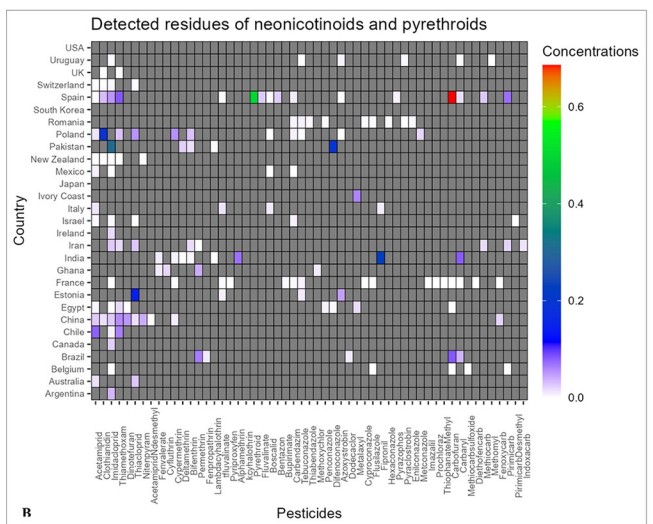

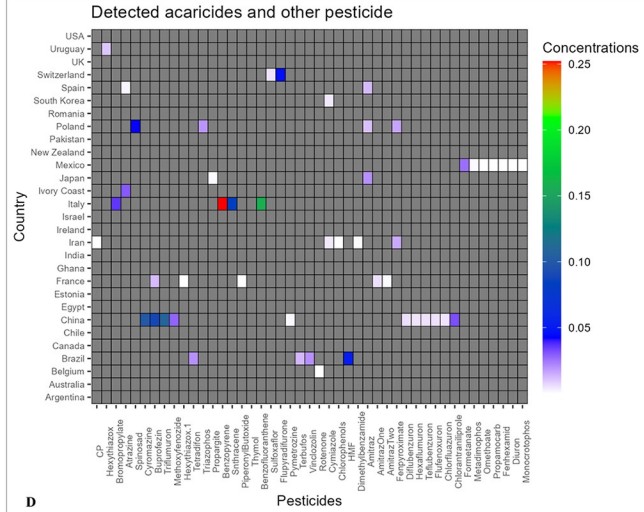

**Fig 5.** Heatmaps with colour scale on the right of each graph showing the detected: A) organochlorines and organophosphates B) neonicotinoids and pyrethroids; C) fungicides and herbicides, and D) acaricides other pesticides which were studied (x-axis) over the period and the respective countries where detections took place (y-axis). Concentrations of each detected pesticide were averaged per the number of detections per country to get one value for each pesticide detected. Units for all detected pesticides were standardised by converting to mg/kg. The individual graphs can be referred to in S2–S5 Figs.

where honey samples were collected and examined multiple times. No pesticide residues were detected in repeated studies conducted in Uganda and Spain. In contrast, in two independent studies conducted in Chile, while no pesticide residues were detected in one study, acetamiprid, thiamethoxam, thiacloprid and imidacloprid were confirmed in three honey samples in the other. In a study conducted in France, where samples were taken from apiaries in the spring, autumn, and early and late summer, contamination was higher in samples taken in the early spring. In a study conducted in Egypt, acetamiprid and imidacloprid were found in honey samples tested in the spring (during the clover season) and summer (during the cotton season). One study in Estonia in 2013 found that the amounts of clopyralid and glyphosate were greater than their designated MRL. However, MRL was not exceeded in different studies conducted in Estonia in 2013 and 2014, where honey samples were taken and analyzed for two years. Frequent pesticide residue detections were found over a nine-year investigation in

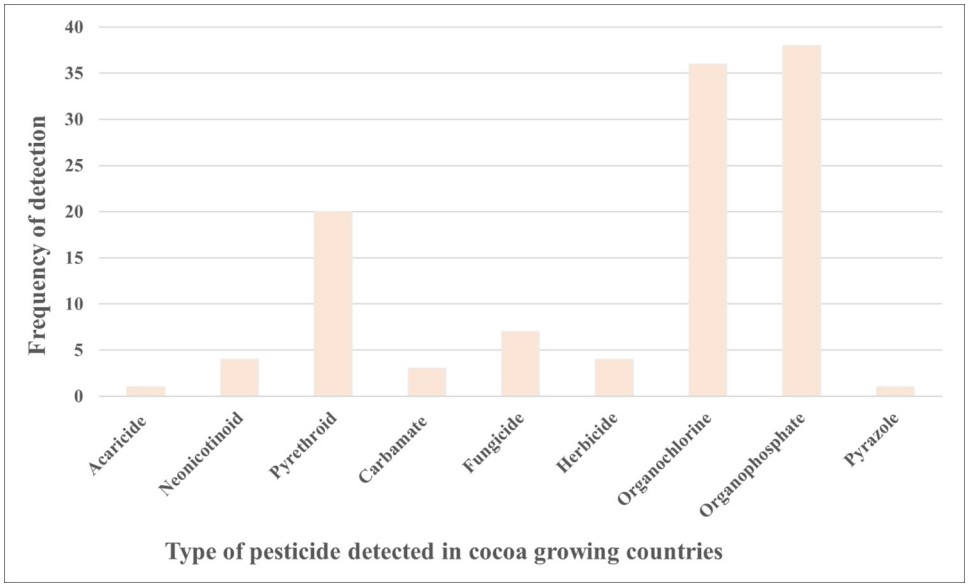

**Fig 6. The frequency of detection of the different classes of pesticide residues in the 20 studies conducted in six cocoa growing countries: Ghana, Ivory Coast, Brazil, Mexico, India, and Pakistan.** No pesticide residues were detected in studies conducted in Thailand and Uganda, the other two cocoa growing countries where studies took place.

Estonia that began in 2004. In a different study, glyphosate was examined in honey samples collected in 2015 and 2016 from two distinct locations in the USA and at both locations, its concentration increased in the 2016 samples.

## 3.5. Limit of detection and limit of quantification applied in studies

The limit of quantification (LOQ) is the smallest chemical concentration that can be successfully quantified [67, 68]. The limit of detection (LOD) is the smallest concentration that can be successfully detected [69, 70]. The LOD and LOQ used for pesticide analysis in each study were assessed (Table 1 and S3 Table). The analysis methods employed in 77% of the included studies resulted in LOQs that fell below the designated EU MRLs for the investigated substances. In these studies, therefore, it was possible to evaluate if the pesticide concentrations detected exceeded the MRL. There was insufficient information in seventeen other cases to determine the LOQ employed. Three of the LOQs that were explicitly mentioned were found to be higher than the MRL. For these 20 studies, therefore, it was not possible to conclude that the absence of a pesticide being detected correlated with these pesticide concentrations not exceeding the relevant MRL.

## 4. Discussion

Honey is beneficial to humankind for its nutritive values and as a medium for monitoring environmental quality by assessing its contents for environmental contaminants. In this present study, we undertook a systematic literature review to evaluate honey contamination from plant protection products recommended for cultivating cocoa (*Theobroma cacao L.*), a crop highly dependent on pesticides for cultivation because of its vulnerability to insect and disease attacks.

　　Our findings demonstrate a steady but low level of analysis of pesticide residues in honey from 1997, with peak reporting periods beginning in 2015. Similar findings were reported by

**Table 3. Concentrations of pesticide residues exceeded the maximum residue limits specified by the European Union by 2022.** Those in bold denote pesticide residues that exceed the previous and revised MRLs. Asterisks denote concentrations that were thought to have exceeded the previous MRL established by the EU in 2005. These concentrations, however, are below the updated MRL that the EU has set as of 2022.

| Pesticide exceeding MRL | Class of pesticide | Concentration mg/kg | Current MRL (mg/kg) | Previous MRL (mg/kg) | Country | Author |
|---|---|---|---|---|---|---|
| Bifenthrin | Pyrethroids | 0.0145 * | 0.05 | 0.01 | Poland | [54] |
| Fenpyroximate | acaricides | 0.0163* | 0.05 | 0.01 | | |
| Methidathion | Organophosphate | 0.0257 * | 0.05 | 0.02 | | |
| Spinosad | | 0.0206* | 0.05 | 0.01 | | |
| Thiamethoxam | Neonicotinoid | 0.0202 * | 0.05 | 0.01 | | |
| Triazophos | Organophosphate | 0.0203 * | 0.05 | 0.01 | | |
| Azoxystrobin | Herbicides | 0.031 * | 0.05 | 0.01 | Estonia | [55] |
| Imidacloprid | Neonicotinoid | **0.55** | 0.05 | 0.05 | Pakistan | [56] |
| Endosulfan | Organochlorine | **0.26** | 0.01 | 0.01 | | |
| Imidacloprid | Neonicotinoid | **0.736** | 0.05 | 0.05 | Pakistan | [57] |
| Difenoconazole | Fungicide | **0.386** | 0.05 | 0.05 | | |
| Bifenthrin | Pyrethroids | **15.76** | 0.05 | 0.01 | | |
| Glyphosate | Herbicides | **2.04** | 0.05 | 0.05 | | |
| Trichlorfon | Organophosphate | **0.029** | 0.01 | - | Brazil | [58] |
| Bifenthrin | Pyrethroids | 0.0172* | 0.05 | 0.01 | Iran | [59] |
| Fenpyroximate | Acaricides | 0.0154* | 0.05 | 0.01 | | |
| Thiamethoxam | Neonicotinoids | 0.0183* | 0.05 | 0.01 | | |
| Tau-flavulinate | Pyrethroids | 0.014* | 0.05 | 0.01 | Italy | [60] |
| Bromopropylate | Acaricide | **0.036** | 0.01 | 0.01 | | |
| Coumaphos | Organochlorine | **0.036** | 0.01 | 0.1 | Spain | [61] |
| Dimethylformamide (DMF) | Amitraz | **0.541** | 0.2 | 0.05 | | |
| DMPF | Amitraz | 0.107* | 0.2 | 0.05 | | |
| Metalaxyl | Fungicide | **0.06** | 0.05 | 0.05 | Ivory Coast | [62] |
| Chlorpyrifos | Metalaxyl | **0.208** | 0.01 | 0.05 | | |
| Atrazine | Herbicide | **0.03** | 0.5 | - | | |
| Glyphosate | Herbicides | **0.22** | 0.05 | 0.05 | Brazil | [63] |

[71], who examined the presence of plant protection product residues in plant pollen and nectar, two sources of raw honey. Most of the research they examined was published from 2015, corresponding with the time we observed a sharp increase in studies investigating pesticide residues in honey. The increased growth in studies after 2014 coincided with when the EU placed a moratorium on using some neonicotinoids, namely clothianidin, imidacloprid and thiamethoxam [72, 73]. Our findings also indicated that most studies conducted in countries where cocoa is grown occurred around this time. It is possible that the sharp growth in studies could be in response to the reported bee deaths due to the pervasive use of pesticides [74] and the reported worldwide decline of pollinators [75]. Notably, the outdoor use of three neonicotinoids—clothianidin, thiamethoxam, and imidacloprid—was made illegal in 2018 in Europe [76, 77]. Studies increased dramatically again in 2022, with the majority in China (Fig 2).

The study's most striking finding is that 19% of studies were conducted in cocoa producing nations, mostly developing countries. This result correlated with the findings that most studies on the impacts of herbicides and fungicides on bees were conducted in North America, Europe and Russia [78]. A similar trend was observed in a study conducted by [71] which evaluated plant protection products in pollen and nectar. Cocoa thrives in hot and humid climatic conditions and tends to flourish in areas around West Africa, East Asia and South America [79]. Accordingly, most cocoa producing countries are located outside North America and Europe.

Our finding highlights a dearth of knowledge of the environmental impact of pesticides imputed for cocoa cultivation. Considering honey as a proxy for such assessments, the paucity of knowledge may restrict a better or more detailed environmental impact assessment. Presently, cocoa production levels do not meet demand in several parts of the world, such as China and India [80], and there is currently an increased 2.5% yearly demand for cocoa beans around the world [81]. This is likely to translate into increased cocoa production with a corresponding increased pesticide use to control disease and insect pests. Prioritising evaluation or studies of honey contamination from pesticide application in cocoa growing areas may help reveal the extent to which honey is impacted by pesticides applied for cocoa cultivation and, by extension, the extent to which these compounds are detectable in these regions.

Neonicotinoids were the most detected pesticide substances evaluated. These results support claims elsewhere ([82–84]) that neonicotinoids are the most widely used class of pesticides globally. Neonicotinoids can persist in woody plants for over 365 days [105], with reported half-lives over 1000 days [83]. Therefore, their detection is possible even many months or years after application. Moreover, since neonicotinoids were developed in the 1980s to replace the more persistent organochlorines in the environment [84, 85], they have been in great demand [86]. Therefore, it was not surprising that imidacloprid, which along with clothianidin is observed to be highly persistent under certain conditions, was the chemical frequently found in this study. This correlates with the findings of [87, 88]. Additionally, as of 2009, imidacloprid had sales of $1091 million, making it the insecticide with the biggest global market share [89]. It is approved for 140 crops, including several crop types such as vegetables, citrus, corn, and oilseed rape pome, among several others, in about 120 countries [89, 90]. It is therefore not surprising that imidacloprid was detected most frequently in our included studies [91, 92].

Even though neonicotinoids were the most detected class of pesticide residues across all research in this study, our findings show that the top three most frequently detected classes of pesticides in the six cocoa-growing countries were organophosphates, organochlorines, and pyrethroids, in that order (Fig 6 and S6 Table). Among the plausible reasons for this finding are that these pesticides are inexpensive and easily accessible and are, therefore, frequently used in developing countries where most cocoa producing countries are located [93–95]. From our study, we can confirm that 60 pesticides, which are largely not approved for cocoa cultivation [50], were detected in studies conducted in cocoa producing countries, though again, it should be noted that it was not possible to uniformly ascertain if the honeys collected came from cocoa producing areas within these countries. Many of these pesticides were found in Mexico and India, for which cocoa production is not the dominant agricultural crop. Implementing laws and regulations governing the use of pesticides in developing countries continues to be a challenge. The ban on using OCPs in developed countries has witnessed remarkable successes [93]. Still, the same may not be vouched for developing countries where pesticides are highly valued as a means of breaking into the global market of food production [96]. Organophosphorus pesticides (OPPs) continue to be widely applied in developing countries due to their ability to inhibit disease attacks and enhance productivity [94].

Three distinct studies conducted in three different nations—Ghana, Mexico, and India—detected pesticides designated as illegal under the Stockholm Convention. However, it is noteworthy that pollutants of organochlorine (OC) derivatives, such as PCBs, DTT, and a number of other pesticides no longer approved for use, have been found to persist in the environment [97]. It was beyond the scope of this work to determine whether the detected illegal pesticides were administered recently or were present in past applications. Even though research findings by Bayoumi [98] point to the continued use of substantial amounts of banned chemical pesticides in developing countries, it must also be recognised that in some countries such as India, DDT, which has received a worldwide ban, is still approved for use against mosquitoes in

controlling malaria [53]. This may explain the frequent detection of DDT and its various derivatives in research carried out in India (S2–S4 Figs and S6 Table).

In the present study, we found that 12% of included studies detected pesticides whose concentrations exceeded allowable limits required for human consumption (Table 3), one quarter of which occurred in three cocoa producing countries. One further important observation from our study shows that some detected pesticide residues that previously exceeded specified MRLs set by the EU at the time of the study are presently below the revised MRLs that have since been implemented in the EU. This is significant as it implies that products previously deemed to pose a risk to human health would now be assessed as not posing an unacceptable risk. It should also be considered that while the revision of MRL can impact the assessment of honey as a food product, it does not alter its relevance as an indicator for the assessment of pesticide contamination in the surroundings of the hive location. The finding of pesticides exceeding MRL is significant in at least two major respects. Human exposure to levels of pesticides exceeding MRL can cause many health-related problems. The consumption of unacceptable levels of pesticides via food is known to have many acute and chronic health implications [99]. Exploration of the causes of exceedances of MRLs is beyond the scope of this work. Nonetheless, it should be noted that as only 0.01% of applied pesticides reach their target, with the rest entering into the general ecosystem, the exceedance of MRLs should serve as a warning for the potential impact of these compounds on the surrounding environment [17, 18].

The LOD and LOQ employed in the bulk of the studies under review were often lower than the MRLs set for honey by the EU, which range from 0.05 mg/kg to 0.2 mg/kg [100]. This finding suggests that studies included in this review largely applied analytical methods with sufficient sensitivity to allow the potential health implications of pesticide detections to be evaluated. It must, however, be noted that the LODs for three studies were not suitable for detecting pesticide residues below EU MRLs, compromising the extent to which their results could be considered within this study. In particular, even though no pesticide residues were found in one Brazilian study [101], their reported LOQs mean that the study's reports of no pesticides detected cannot expressly be interpreted to mean that no pesticides were present at concentrations that could cause harm. In the studies by [102, 103], the LOQs attained for the method were at concentrations so high that their results cannot be construed to suggest that the pesticides detected were the only ones that were of concern.

Even though the scope of this study did not extend to assessing the effects of pesticides on bees, the high frequency of detection of neonicotinoids in honeys as observed in our study suggests there is a potential risk that bees could be impacted by neonicotinoids through exposure during foraging. In our study, concentrations of 0.736 mg/kg of imidacloprid, [56]; 0.0274 mg/kg of thiacloprid [104] and 0.0202 mg/kg of thiamethoxam [54] were confirmed in Pakistan and Poland respectively. Although these concentrations are below the known $LD_{50}$ for these compounds for bees [105], they are within the range of concentrations shown to induce sublethal effects. For instance, [106] confirmed that the survival of honeybees was reduced by 51% when exposed to 0.0043 mg/kg and 0.0011 mg/kg concentrations of thiamethoxam and clothianidin respectively. Brood development was stunted when honey bees were exposed to field-realistic concentrations of thiamethoxam (0.2 mg/kg) and clothianidin (0.001 and 0.01 mg/kg) [107]. Bumble bees were found to experience reduced learning capability, and have changes in foraging and homing success, when exposed to field realistic levels of up to 0.0024 mg/kg of thiamathoxam [108], which is 10-fold lower concentration than what was detected by Bargańska et al. in Poland. Therefore, the possibility of sub-lethal effects of the detected pesticide residues on honeybees should not be ruled out.

It was observed that pesticide residues were detected in 80% of commercial and raw honeys analysed in the included studies. This discovery is consistent with the findings of [87], who, in

a global study of neonicotinoids in honeys, verified the presence of neonicotinoids in 75% of 198 honeys obtained directly from producers. However, the most striking observation made in our study was that 90% of studies that analysed raw honeys confirmed the presence of pesticide residues. This was higher than previous findings [87] evaluating raw honeys. However, it should be noted that our findings were not confined to neonicotinoids. Our finding highlights the frequent occurrences of pesticides in the general environment. Raw honeys from a broad spectrum of natural and agricultural landscapes were assessed in the studies of interest in this review. These included raw honeys from agricultural farmlands within forest belts in Ghana [109], apiaries located within 2 miles of an oilseed [110], various agroclimatic zones [111], agricultural landscapes with mostly intensively managed fields, forested areas and human settlements [55], unifloral and multifloral sources [112] among several others. In the present study, a very small number of studies evaluated the floral background of the honey. Therefore, it was not possible to correlate pesticide contamination to any specific floral resources.

## 5. Conclusion

The current knowledge of studies of honey contamination from pesticides approved for cocoa cultivation has been evaluated through a systematic literature review. The studies conducted to date have been disproportionately focused on non-cocoa growing countries, leaving a huge gap in knowledge of what residues of pesticides approved for cocoa cultivation are found in honey and, by proxy, how prevalent these pesticides are in the environment in cocoa growing areas. Future research should therefore prioritize cocoa producing nations, particularly the top producers, Ghana, and Ivory Coast, who together produce 70% of the world's cocoa. Continuous monitoring and rigorous adherence to pesticide application regulations are crucial in cocoa production to ensure pesticide residues are kept below harmful levels. Using analytical techniques with appropriate sensitivity, stakeholders can ensure that residue levels can be evaluated using MRLs to minimise potential negative impacts. Outcomes from these studies could contribute to policy formulation of pesticide usage, human health, and sustainable beekeeping, especially in cocoa production landscapes.

## Supporting information

**S1 Appendix. The scoring scheme applied to evaluate the quality of included studies (Adapted from [47]).** This system enabled a standardized and rigorous appraisal of research methodology and relevance, enhancing the overall reliability of the study findings.
(DOCX)

**S1 Fig. The results of the updated data search performed to retrieved additional studies conducted between November 2020 and November, 2022.** Sixteen 16 publications were retrieved and subsequently added bringing the total papers to 104.
(TIF)

**S2 Fig. A heatmap with a colour scale on the right, illustrating the presence of studied organochlorines and organophosphates over time (x-axis) across various countries (y-axis).** The concentrations of each detected pesticide were averaged per country to generate a single value for each pesticide in each country for the purpose of this visualization.
(TIF)

**S3 Fig. A heatmap featuring a colour scale on the right, portraying the presence of neonicotinoids, pyrethroids, and carbamates (x-axis) studied over time, and their corresponding occurrences in various countries (y-axis).**
(TIF)

**S4 Fig. A heatmap with a colour scale on the right, depicting the analysis of fungicides and herbicides (x-axis) across time, and their occurrences in different countries (y-axis).**
(TIF)

**S5 Fig. A heat map, accompanied by a colour scale on the right, illustrating the exploration of acaricides and other pesticides (x-axis) across time and their identification in various countries (y-axis).**
(TIF)

**S1 Table. Search strings used to retrieve articles from search engines.** These search strings were meticulously crafted based on the key pesticides relevant to cocoa cultivation. The selection of these pesticides encompassed twenty-three insecticides, seventeen fungicides, and two herbicides, all of which have been approved for use in cocoa growing.
(DOCX)

**S2 Table. The customized checklist designed to evaluate and appraise the papers that met the specified inclusion criteria.** This carefully tailored checklist served as a comprehensive tool to assess the selected papers in a structured manner, ensuring that they align with the predefined criteria for inclusion and covered.
(XLSX)

**S3 Table. Dataset comprising the list of 104 papers which satisfied both inclusion criteria and quality assessment.** This encompasses papers retrieved from both the original search and the subsequent updated search.
(XLSX)

**S4 Table. Detailed inventory of the included papers, featuring their respective aims, study designs, key findings, summarized content, and key results.**
(XLSX)

**S5 Table. Cocoa-producing countries categorized by the metric tons of cocoa beans they have produced.** Notably, studies were conducted only in eight cocoa-producing countries.
(DOCX)

**S6 Table. The number of pesticide residues that were detected across different studies conducted in cocoa growing countries.**
(XLSX)

**S7 Table. An overview of the number and concentrations of pesticide residues detected in countries where multiple studies took place.** Generally, the concentrations of these pesticides exhibited fluctuations in different studies. It was only in Estonia where azoxystrobin was detected at an identical concentration in two separate different studies.
(DOCX)

**S8 Table. PRISMA checklist.**
(DOCX)

## Acknowledgments

We thank Helen Sheridan and Thomas Cummins for their valuable inputs and discussion. Linzi J. Thompson, Elena Zioga and Diarmuid Stokes are also acknowledged for their useful discussion on conducting a systematic literature review. We would like to acknowledge the

support provided by the IReL Open Access agreement, which has facilitated the accessibility and dissemination of our research findings.

## Author Contributions

**Conceptualization:** Richard G. Boakye, Dara A. Stanley, Blanaid White.

**Data curation:** Richard G. Boakye, Blanaid White.

**Formal analysis:** Richard G. Boakye, Dara A. Stanley, Blanaid White.

**Funding acquisition:** Richard G. Boakye, Dara A. Stanley.

**Investigation:** Richard G. Boakye, Dara A. Stanley, Blanaid White.

**Methodology:** Richard G. Boakye, Dara A. Stanley, Blanaid White.

**Project administration:** Richard G. Boakye, Dara A. Stanley, Blanaid White.

**Resources:** Richard G. Boakye, Dara A. Stanley, Blanaid White.

**Supervision:** Dara A. Stanley, Blanaid White.

**Validation:** Richard G. Boakye, Dara A. Stanley, Blanaid White.

**Visualization:** Richard G. Boakye, Dara A. Stanley, Blanaid White.

**Writing – original draft:** Richard G. Boakye, Blanaid White.

**Writing – review & editing:** Richard G. Boakye, Dara A. Stanley, Blanaid White.

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
