## [Decision Letter · Decision Letter 0]

19 Apr 2023

PONE-D-22-31708Honey contamination from plant protection products approved for cocoa cultivation: a  systematic review of existing research and methodsPLOS ONE

Dear Dr. Boakye,

Thank you for submitting your manuscript to PLOS ONE. After careful consideration, we feel that it has merit but does not fully meet PLOS ONE’s publication criteria as it currently stands. Therefore, we invite you to submit a revised version of the manuscript that addresses the points raised during the review process.

We look forward to receiving your revised manuscript.

Kind regards,

Javaid Iqbal, PhD

Academic Editor

PLOS ONE

2. We note that [Figures 3 and 4] in your submission contain [map/satellite] images which may be copyrighted. All PLOS content is published under the Creative Commons Attribution License (CC BY 4.0), which means that the manuscript, images, and Supporting Information files will be freely available online, and any third party is permitted to access, download, copy, distribute, and use these materials in any way, even commercially, with proper attribution. For these reasons, we cannot publish previously copyrighted maps or satellite images created using proprietary data, such as Google software (Google Maps, Street View, and Earth). For more information, see our copyright guidelines: http://journals.plos.org/plosone/s/licenses-and-copyright.

a. You may seek permission from the original copyright holder of Figures 3 and 4 to publish the content specifically under the CC BY 4.0 license.  

Natural Earth (public domain): http://www.naturalearthdata.com/.

3. Please include your tables as part of your main manuscript and remove the individual files. Please note that supplementary tables (should remain/ be uploaded) as separate ""supporting information"" files.

Reviewers' comments:

Reviewer's Responses to Questions

**Comments to the Author**

1. Is the manuscript technically sound, and do the data support the conclusions?

Reviewer #1: Partly

Reviewer #2: Yes

2. Has the statistical analysis been performed appropriately and rigorously? 

Reviewer #1: N/A

Reviewer #2: I Don't Know

3. Have the authors made all data underlying the findings in their manuscript fully available?

Reviewer #1: Yes

Reviewer #2: Yes

4. Is the manuscript presented in an intelligible fashion and written in standard English?

Reviewer #1: Yes

Reviewer #2: No

5. Review Comments to the Author

Reviewer #1: The manuscript needs condensation.

Very long flavorless sentences are found.

Exaggeration is common.

Inclusive results and discussions are needed.

The overall manuscript should be reorganized.

Data, especially results are not ordered.

Some figures are traditional.

Data of MRLs should be organized in a table.

Reviewer #2: 1- There are multiple gaps in the references which must be corrected.

2- The main body of the article needs to be revised for English language.

3- There is no mention of pesticides approved for cacao cultivation in the MS.

4- page 28: Since the amount and type of pesticides used vary from one region to another, how can a specific standard for MRL be determined across borders?

5- page 31: It is a good idea to consider honey as an indicator of monitoring pesticides. But,In the leading and exporting countries of cocoa, isn't it better to analyze the residues of pesticides used, on the cocoa plant itself?

6. PLOS authors have the option to publish the peer review history of their article (what does this mean?). If published, this will include your full peer review and any attached files.

Reviewer #1: **Yes: **Ayman A. Owayss

Reviewer #2: No

---

## [Author Response · Author response to Decision Letter 0]

23 Jun 2023

Responses to Editor’s Comments

Editor’s Remarks

Editor’s remark 1. Please ensure that your manuscript meets PLOS ONE's style requirements, including those for file naming. 

Responses: The work has now been formatted in accordance with PLOS ONE's style guidelines using the suggested template.

Editor’s remark 2. We note that [Figures 3 and 4] in your submission contain [map/satellite] images which may be images which may be copyrighted.

Responses: Thank you for highlighting this concern. The maps were drawn in Excel using Geonames. The Copyright requirements of Geonames (https://www.geonames.org/) are compatible with the Creative Commons Attribution CC BY 4.0 license https://creativecommons.org/licenses/by/4.0/) which allows copying, redistribution and transformation of material in any medium or format so long as due recognition is given. In the manuscript previously submitted, the images were not given due acknowledgement or attributed to Geonames. We have updated the captions of the figures to attribute recognition in this revision.

Editor’s remark 3. Please include your tables as part of your main manuscript and remove the individual files. Please note that supplementary tables (should remain/ be uploaded) as separate ""supporting information"" files.

Response: We appreciate your advice. The three tables have been added where necessary and have been deleted from the files in accordance with your instructions. The data for the MRLs that reviewers suggested we include is already in Table 3. We wish to indicate that tables designated as supplementary have been submitted and kept separate as requested.

Reviewer's Responses to Questions & Comments

1. Is the manuscript technically sound, and do the data support the conclusions? The manuscript must describe a technically sound piece of scientific research with data that supports the conclusions. Experiments must have been conducted rigorously, with appropriate controls, replication, and sample sizes. The conclusions must be drawn appropriately based on the data presented.

Reviewer #1: Partly 

Reviewer #2: Yes

Response: This study was primarily a literature review and therefore neither field nor lab experiments were necessary. 

2. Has the statistical analysis been performed appropriately and rigorously?

Reviewer #1: N/A

Reviewer #2: I Don't Know

Response: We appreciate your inquiry on whether a suitable and sufficient statistical analysis was performed. Statistical analyses were neither applicable nor necessary in this research, as one reviewer noted. Our study's design eliminates the necessity for statistical analysis. Nevertheless, thorough quantitative and qualitative analyses were carried out, and they are presented in our review with support from the provided tables, figures, and narratives.

Editor’s 3. Have the authors made all data underlying the findings in their manuscript fully available?

Response: Both the reviewers gave 2 Yes so no further action was carried based on their responses.

4. Is the manuscript presented in an intelligible fashion and written in standard English?

Reviewer #1: Yes

Reviewer #2: No

Response: Because the language and wording of the book did not receive unanimous approval from the two reviewers, the manuscript has been reviewed in detail with the manuscript wording revised as appropriate to meet the standards of both the reviewers and PLOS ONE. We are certain that any ambiguity or grammatical errors that previously existed have been properly corrected by this version.

Authors response to Reviewer 1: 

Comment 1 of reviewer 1 - Very long flavourless sentences are found.

Response: We have read through the manuscript and edited any phrases that are excessively long

Comment 2 of reviewer 1 - Exaggeration is common.

Response: We have reviewed the manuscript in detail and revised as appropriate to ensure that all analysis and discussion is accurate and not exaggerated. In its current form, we are certain that overblown points have been reduced to accurately communicate our study's true conclusions.

Comment 3 of reviewer 1 - The overall manuscript should be reorganized.

Response: A number of sections have been reorganised including Section 3.2 which was formerly section 3.5. The period and geographical regions where studies are currently being conducted are smoothly transitioning into the various insecticides used in those regions. 

Comment 4 of reviewer 1 - Data, especially results are not ordered.

Response: We extensively address this comment in our response to reviewer 1's Comment 3. As previously mentioned, part 3.2, which is now section 3.5, formerly interrupted the flow of the results presentation. Sections 3.1 in this updated version discuss the time frame and locations where the experiments were conducted. Following this, sections 2 and 3 list every pesticide that was found throughout the period and all those that exceed the MRL, along with an updated table. The types of honey that were analyzed and the sampling frequency are then highlighted in Section 4. Then, Sections 5 and 6 cover all topics relating to methodology. We think the new version's result section's data is thoughtfully organized

Comment 5 of reviewer 1 - Some figures are traditional.

Response: Changes have been made to Figures 2 and 6. For better comparison of the frequency of detections, figure 6, in particular, has been converted from pie chat to bar chat.

Comment 6 of reviewer 1 - Data of MRLs should be organized in a table

Response: We are aware of the reviewer's request for the MRL data to be arranged in a table. We would like to point out that Table 3—which was provided as a separate attachment in accordance with the submission guidelines—already contains the data for the MRLs. However, we have included it in the manuscript as the editors requested, and we anticipate that it will satisfy the reviewer's comment. 

Comment 7 of reviewer 1 - The manuscript needs condensation.

Response: The manuscript has been reviewed and condensed by approximately 20%. 

Comment 8 of reviewer 1 - Inclusive results and discussions are needed.

Response: Data extracted from the included studies based on Table 1, designed to guide data extraction, have fully been presented and given sufficient discussions. In response to reviewer’s comments, results and discussions have been re-evaluated and updated accordingly. 

Authors response to Reviewer 2: 

Comment 1 of reviewer 2- There are multiple gaps in the references which must be corrected.

Response: We appreciate you highlighting this. We have updated the document to fix all identified reference gaps. 

Comment 2 of reviewer 2- The main body of the article needs to be revised for English language.

Response: We reviewed the entire text in detail and corrected any sentences that we believed needed work in response to reviewer 2's comments. Before looking more closely at the remaining sentences, we initially focused on the areas that reviewers had called attention to. In the amended manuscript, all areas that have been changed are highlighted with track modifications. 

Comment 3 of reviewer 2- There is no mention of pesticides approved for cacao cultivation in the MS.

Response: The reviewer’s comment is helpful, because it has highlighted for us that we should bring the list of pesticides approved for cocoa cultivation from supplementary information into the manuscript itself. This has now been included in the revised manuscript. This should aid the reader when we then discuss specific compounds on this list throughout the manuscript. Examples of where this list is utilised include in section 3.2, where it is stated, "Eleven approved insecticides for cocoa cultivation, namely capsaicin, chlorantraniliprole, thiamethoxam, acetaprimid, etofenprox, indoxacarb, pirimiphosmethyl, promecarb, pyrethrum, sulfoxaflor, and teflubenzuron, and one herbicide. In addition, we said on page 6 that there were 2 herbicides, 23 insecticides, and 17 fungicides on the list of chemicals authorized for use in cocoa farming. It should also aid the reader in their understanding of the heatmaps in figure 5. Both the pesticides authorized for use in cocoa farming and every other pesticide found throughout the study period covered by all publications are shown in Figure 5 and Figures S1 to S4. 

Comment 4 of reviewer 2- page 28: Since the amount and type of pesticides used vary from one region to another, how can a specific standard for MRL be determined across borders?

Response: We acknowledge the reviewer's worry regarding our study's use of a single standard—more particularly, the EU MRL—to evaluate MRL across international borders and think this concern is well-founded. First, while pesticides vary, the residues that can affect human health do not change, and given that the EU is such a large market even foods grown in different countries and regions still need to meet these MRL levels for sale in the EU. The FAO established the Codex Alimentarius in 1963 with the mandate to establish safety standards for all foods and animal feed in order to ensure their suitability for human consumption. In determining MRLs for its regional block, the European Union, one of the largest regional market blocks, considers both Codex Alimentarius requirements and good agricultural practices. Additionally, the EU MRL was employed as the standard of measurement in more than 95% of the research we analyzed for this analysis. As a result, the EU MRL was chosen for our study because it was already being used by the bulk of studies. The concentrations of identified pesticides indicated in various units were converted to milligrams of residue per kilogram of feed commodity (mg/kg), which is used in the EU, to ensure direct comparison. It was simple to compare units across borders after all the units were standardized. The fact that different countries' usage can be a potential source of misunderstanding was another important conclusion that emerged from our research, and for that reason we advocate for the necessity of standardizing MRL units across national borders.

Comment 5 of reviewer 2- page 31: It is a good idea to consider honey as an indicator of monitoring pesticides. But In the leading and exporting countries of cocoa, isn't it better to analyse the residues of pesticides used, on the cocoa plant itself?

Response: This is a really interesting question from the reviewer. The short answer is that it depends, and on multiple factors, including the purpose of the study, the availability of samples for analysis (particularly for retrospective analysis), and the availability of benchmarking data. In our case, the goal of our review was to gauge environmental pesticide contamination in cocoa growing countries. Honey was chosen for our investigation because it has been widely as a proxy for environmental monitoring, and accepted for use as such. Due to the foraging behaviour of bees, pesticide residues found in honey may be interpreted as a reflection of pesticides used in the area surrounding the hive and, thus, as a reflection of the environment's overall health. Additionally, due to honey’s chemicophysical properties, compounds incorporated tend to be stable, allowing for retrospective analysis. In addition to the fact that honey may be used to measure the levels of environmental pollution, the evaluation of pollutants in honey can also reveal if the honey is safe for ingestion by people. Additionally, herbicides have been named as one of the factors contributing to a global drop of pollinators. Therefore, it is possible to deduce the potential consequences of pesticides found in honey on honeybees. The possibility of excess pesticide applications to cocoa affecting the general environment should be a concern because studies have shown that only about 0.01% of applied pesticides are determined to reach their target with the remaining filtering into general ecosystem. For this reason, using pesticide residues in honey to measure environmental contamination makes sense.

---

## [Decision Letter · Decision Letter 1]

7 Aug 2023

Honey contamination from plant protection products approved for cocoa (Theobroma cacao ) cultivation: A systematic review of existing research and methods

PONE-D-22-31708R1

Dear Dr. Boakye,

We’re pleased to inform you that your manuscript has been judged scientifically suitable for publication and will be formally accepted for publication once it meets all outstanding technical requirements.

Kind regards,

Charles Odilichukwu R. Okpala

Academic Editor

PLOS ONE

Additional Editor Comments (optional):

Thank you authors for diligently revising your work.

Reviewers have checked it and the editor agrees with their recommendation to accept it for publication.

Thank you for finding PlosONE as your journal of choice.

Look forward to your future scholarly contributions

Reviewers' comments:

Reviewer's Responses to Questions

**Comments to the Author**

1. If the authors have adequately addressed your comments raised in a previous round of review and you feel that this manuscript is now acceptable for publication, you may indicate that here to bypass the “Comments to the Author” section, enter your conflict of interest statement in the “Confidential to Editor” section, and submit your "Accept" recommendation.

Reviewer #1: All comments have been addressed

Reviewer #3: All comments have been addressed

2. Is the manuscript technically sound, and do the data support the conclusions?

Reviewer #1: Yes

Reviewer #3: Yes

3. Has the statistical analysis been performed appropriately and rigorously? 

Reviewer #1: (No Response)

Reviewer #3: N/A

4. Have the authors made all data underlying the findings in their manuscript fully available?

Reviewer #1: Yes

Reviewer #3: Yes

5. Is the manuscript presented in an intelligible fashion and written in standard English?

Reviewer #1: (No Response)

Reviewer #3: Yes

6. Review Comments to the Author

Reviewer #1: None.............

..................

.....

.

.....

.........................................

..

Reviewer #3: The authors carried out a study on Honey contamination from plant protection products approved for cocoa (Theobroma cacao) cultivation: A systematic review of existing research and methods. This is an article of interest in the field of science.

1. The topic is unique and worthy of researching.

2. The abstract is informative and do reflect the body of the paper.

3. The introduction provides sufficient background information for readers in the immediate field to understand the problem/hypotheses.

4. The text is well arranged, and the logic is clear. The related concepts were introduced clearly. The readability is sufficient.

5. The proposed simulation /scheme was quite novel

6. The theoretical analysis in this article is strong.

7. All figures/tables are clear enough to summarize the results for presentation to the readers. All figures/tables are well referred to in the text.

8. The deduced conclusions were based on the research methods/cases

9. The conclusions are tenable. With these conclusions, much progress has been made when compared with the existing research findings.

10. The reference section is informative and accurate.

7. PLOS authors have the option to publish the peer review history of their article (what does this mean?). If published, this will include your full peer review and any attached files.

Reviewer #1: **Yes: **Ayman A Owayss

Reviewer #3: **Yes: **Iheanyi Omezuruike Okonko

---

## [Editor Report · Acceptance letter]

2 Oct 2023

PONE-D-22-31708R1 

Honey contamination from plant protection products approved for cocoa (*Theobroma cacao* ) cultivation: A systematic review of existing research and methods 

Dear Dr. Boakye:

I'm pleased to inform you that your manuscript has been deemed suitable for publication in PLOS ONE. Congratulations! Your manuscript is now with our production department. 

Kind regards, 

on behalf of

Dr. Charles Odilichukwu R. Okpala 

Academic Editor

PLOS ONE